# 2021 Update on Diagnostic Markers and Translocation in Salivary Gland Tumors

**DOI:** 10.3390/ijms22136771

**Published:** 2021-06-24

**Authors:** Malin Tordis Meyer, Christoph Watermann, Thomas Dreyer, Süleyman Ergün, Srikanth Karnati

**Affiliations:** 1Department of Otorhinolaryngology, Head and Neck Surgery, University of Giessen, Klinikstrasse 33, Ebene -1, 35392 Giessen, Germany; malinmeyer@hotmail.de (M.T.M.); Christoph.Watermann@chiru.med.uni-giessen.de (C.W.); 2Institute for Pathology, Justus Liebig University, Langhansstrasse 10, 35392 Gießen, Germany; th.dreyer@gmx.de; 3Institute for Anatomy and Cell Biology, Julius-Maximilians-University Würzburg, Koellikerstrasse 6, 97070 Würzburg, Germany; sueleyman.erguen@uni-wuerzburg.de

**Keywords:** salivary gland tumors, epithelial salivary gland, adenoid cystic carcinoma (ACC), pleomorphic adenoma, mucoepidermoid carcinoma, diagnostic markers

## Abstract

Salivary gland tumors are a rare tumor entity within malignant tumors of all tissues. The most common are malignant mucoepidermoid carcinoma, adenoid cystic carcinoma, and acinic cell carcinoma. Pleomorphic adenoma is the most recurrent form of benign salivary gland tumor. Due to their low incidence rates and complex histological patterns, they are difficult to diagnose accurately. Malignant tumors of the salivary glands are challenging in terms of differentiation because of their variability in histochemistry and translocations. Therefore, the primary goal of the study was to review the current literature to identify the recent developments in histochemical diagnostics and translocations for differentiating salivary gland tumors.

## 1. Introduction

Salivary gland tumors are rare neoplasias of the head and neck that have an annual incidence of 7.03 to 8.58/100,000 [1]. The prevalence of salivary gland cancer is even lower, with 16/1,000,000 [2]. There are 33 existing tumor entities of epithelial origin, and 2–5% of the tumors are non-epithelial. Based on the known entities, it can be highlighted that epithelial salivary gland tumors consist of 10 benign and 23 malignant subtypes, which are the most commonly prevailing in daily clinical life [3]. The primary problem lies in diagnosing a salivary gland tumor, as it is a relatively uncommon form of lesion with wide variations in histological and biological aspects [4]. These tumors have a significant morphological diversity with multiple overlapping features and low incidence rates [5]. The research also elaborates the present situation of the scientific research in immunohistochemistry with paraffin-embedded slides and translocation with fluorescence in situ hybridization (FISH).

### 1.1. Histology of the Healthy Salivary Gland

The salivary glands can be divided into two groups: the minor and major salivary glands—the latter consisting of the parotid, the submandibular, and the sublingual glands. However, the origins of the glands vary. The parotid gland is derived from ectoderm in the sixth and the seventh week along with the submandibular gland. In addition, the submandibular and sublingual glands originate from endoderm. The sublingual gland develops two weeks later than the other two in the ninth week. Furthermore, the parotid gland is considered to be a pure serous gland, while the submandibular gland produces mixed, predominantly serous fluid. On the other hand, the sublingual gland is a mixed, predominantly mucous gland, and the minor salivary glands produce seromucous fluid [6].

The histology of the salivary glands can be categorized into two compartments: luminal (acinar and ductal cells) and abluminal cells (myoepithelial and basal cells). Ideally, routine diagnosis of salivary gland tumors is carried out by hematoxylin–eosin (HE) staining, and immunohistochemistry is preferred to enhance the accuracy of the analysis. A normal salivary gland reacts positively to CEA, EMA, gross cystic disease fluid protein-15, Pan-cytokeratin (CK) (AE1/AE3), mitochondrial markers, alpha-amylase, CK14, p63, calponin, MSA, SMA, and vimentin. However, the reaction with Ki-67 (MIB-1), S-100, and HER2 varies, and androgen receptor (AR), EBER in situ hybridization, lymphoid cell markers, Melan A, p53, and the renal cell carcinoma marker CD10 react negatively [7]. Table 1 presents various studies comprising various cell types, the markers used, and the reaction of these markers.

### 1.2. Current Status of Salivary Gland Tumors and Markers

Salivary gland cancer is categorized as head and neck cancer and constitutes most head and neck cancer diagnoses. Salivary gland tumors differ in glandular cell type, including morphological diversity. Thus, these tumors arise from either the major or several minor salivary glands. Parotid gland tumors are rare and are characterized by heterogeneous entities. The morphological diversity of such tumors and their rarity make them challenging to diagnose. Another factor that adds to the complexity of diagnosis is their subtypes. Diagnosis based on hematoxylin and eosin staining procedures remains the gold standard of salivary gland pathology. Immunohistochemistry combined with hematoxylin and eosin staining can be more useful in specific applications. Ki-67 has the prognostic ability and the potential to differentiate between benign basal cell adenomas and malignant basal cell adenocarcinomas. Similarly, CD43 and Cyclin-A are beneficial for diagnosing adenoid cystic carcinomas and carcinoma ex pleomorphic adenomas. In particular, beta-catenin has two significant functionalities—namely cell–cell adhesion and transcription activation—for various genes that are used for controlling cell proliferation. Platelet-derived growth factor (PDGF) comprises a group of polypeptides responsible for cell proliferation and chemotaxis. Additionally, cyclooxygenase (Cox) enzymes are responsible for catalyzing the synthesis of prostanoids in arachidonic acid metabolism. A study revealed that CEA, Cox-1, Cox-2, PGDF-beta, and WISP-1 and β-catenin are potential diagnostic markers for differentiating benign and malignant parotid gland lesions [8]. This emphasizes that diagnostic markers vary in terms of the type of salivary gland tumor and are used in different forms for accurate diagnosis.

#### 1.2.1. Acinar and Adenoid Cell Carcinomas

An adenoid cystic carcinoma (ADCC) is a malignant form of biphasic epithelial tumor comprising myoepithelial and ductal cells with a high recurrence rate. It is most often encountered at 50 to 70 years of age in the central and minor salivary glands [9]. ADCC constitutes 4.4% of all salivary gland tumors and 11.8% of salivary gland malignancies [9]. These tumors have different patterns such as cribriform, tubular, and solid patterns and mixed forms of growth patterns within the same lesion. Adenoid cystic carcinomas also express ductal and myoepithelial/basal cell markers such as CK7, CAM 5.2, calponin, p63, SOX10, S100, and SMA. The acinar or ductal epithelial cells are generally positive for keratins (CK7 and CAM 5.2) and epithelial membrane antigen (EMA). Usually, they tend to be focally positive or negative for high-molecular-weight keratins (HMWK, CK5/6, and 34BE12). Additionally, they are negative for p63, myoid markers (smooth muscle myosin heavy chain (SMMHC), smooth muscle actin (SMA), calponin), and CK20 (weak focal expression can be observed in rare salivary gland carcinomas). However, myoepithelial cells were found to be positive for p63, myoid markers (SMMHC, SMA, calponin), vimentin, S100, and HMWKs (CK5/6, 34BE12), while they show a weak expression for CK7 and CAM 5.2 and no expression for EMA [10]. ADCCs react positively with calponin, CD43, CK7, CK8, CK14, CK17, CK19, C-Kit, DOG-1, KIT, Maspin, Mcl-1, MCM2, Mit, MYB, NM23, p63, p73, S-100, SMA, SMM, SOX4, and SOX10. They sometimes react positively to caldesmon and GFAP. Apocrine, carbohydrate Ag19-9, carbonic anhydrase VI, CEA, CD9, CK20, HMGA-2, LPLUNC1, PLAG1, SPLUNC1, and SPLUNC2 usually react negatively.

Table 2 and Table 3 present a summary of the literature in terms of adenoid cell carcinomas (ACCs) and acinic cell carcinomas and the corresponding reactions of the patients’ probes to the markers.

Acinic cell carcinomas (ACCs) demonstrate serous acinar differentiation alongside intercalated ductal epithelial differentiation. These tumors have various growth patterns: solid/lobular, microcystic, papillary cystic, and follicular. It is more frequently diagnosed in women (58.8%) than in men (41.2%) [29]. Acinic cell carcinomas usually express CK7 and CAM 5.2. In addition, normal acinar cells express amylase, and most acinic cell carcinomas are found to be negative for this marker. Most reactions of ACC with alpha-1-antitrypsin, carbonic anhydrase VI, chymotrypsin, CK8, CK19, DOG-1, KIT, MCM2, p53, p73, SOX10, synaptophysin, and vimentin are positive. Apocrine, maspin, Mit, and S-100 sometimes react negatively. For amylase, caldesmon, calponin, chromogranin, CK14, CK17, CK20, GFAP, HMGA-2, LPLUNC1, MYB, p63, PLAG1, SMA, SMM, SPLUNC1, and SPLUNC2, adverse effects were observed.

**Table 3 ijms-22-06771-t003:** Analysis of results for acinic cell carcinoma patient probes.

Study	Sample Size ACC	Marker	Reaction	in %
[30]	Six	Alpha-1-Antitrypsin	3 of 6	50%
[31]	Three of parotid origin	Amylase	0 of 3	0%
[11]	Four of parotid origin	Apocrine	1 of 4	25%
[11]	Four of parotid origin	Caldesmon	0 of 4	0%
[11,12]	Eight of salivary gland and head and neck origin; four of parotid origin	Calponin	0 of 8; 0 of 4	0%
[14]	28 of salivary gland origin	Carbonic anhydrase VI (CA6)	28 of 28	100%
[30]	Six	Chromogranin	0 of 6	0%
[30]	Six	Chymotrypsin	4 of 6	67%
[11]	Four of parotid gland origin	CK7	4 of 4	100%
[11]	Four of parotid origin	CK 8	4 of 4	100%
[11]	Four of parotid origin	CK14	0 of 4	0%//+
[11]	Four of parotid origin	CK 17	0 of 4	0%//+
[11]	Four of parotid origin	CK19	4 of 4	100%
[11]	Four of parotid origin	CK20	0 of 4	0%
[14,18]	28 of salivary gland origin; 28 of salivary gland origin	DOG-1	28 of 28; 28 of 28	100%; 100%
[12]	Eight of salivary gland and head and neck origin	GFAP	0 of 8	0%
[19]	One of salivary gland origin or potential mimickers	HMGA-2	0 of 1	0%
[21]	Seven of upper aerodigestive tract origin	KIT (CD117)	6 of 7	86%
[22]	Nine of salivary gland origin	LPLUNC1	0 of 9	0%
[23,24]	Five of salivary gland origin; eleven of salivary gland origin	Maspin	5 of 5; 0 of 11	100%; 0%
[23]	Five of salivary gland origin	MCM2	5 of 5	100%
[11]	Four of parotid origin	Mit	1 of 4	25%
[21]	Seven of upper aerodigestive tract origin	MYB	0 of 7	0%
[13]	Five	p53	2 of 5	40%
[12,27]	Eight of salivary gland and head and neck origin; eight	p63	0 of 8; 0 of 8	0%; 0%
[27]	Eight	p73	5 of 8	63%
[19]	One of salivary gland origin or potential mimickers	PLAG1	0 of 1	0%
[12,14]	Eight of salivary gland and head and neck origin; 28 of salivary gland origin	S-100	1 of 8; 2 of 28	13%; 7.1%
[11,12]	Eight of salivary gland and head and neck origin; four of parotid origin	SMA	0 of 8; 0 of 4	0%
[11]	Four of parotid origin	SMM	0 of 4	0%
[22]	Nine of salivary gland origin	SPLUNC1	0 of 9	0%
[22]	Nine of salivary gland origin	SPLUNC2	0 of 9	0%
[12]	Eight of salivary gland and head and neck origin	SOX10	8 of 8	100%
[30]	Six	Synaptophysin	4 of 6	67%
[14]	28 of salivary gland origin	Vimentin	23 of 28	82%

#### 1.2.2. Mucoepidermoid Carcinomas

Mucoepidermoid carcinomas (MECs) are malignant epithelial neoplasms composed of mucous, epidermoid, intermediate columnar, clear, and oncolytic cells. Approximately 26% of malignant salivary gland tumors are mucoepidermoid carcinoma [1]. It is usually discovered in the fifth decade of life with a slightly elevated prevalence in the female population [32]. Recently, it was discovered that early-stage MECs may grow predominantly intracystically and are thus easier to resect. This assumption emphasizes the importance of an early diagnosis for a good outcome [33]. MECs appear positive for CK5, CK6, CK7, CK8, CK14, CK18, CK19, EMA, CEA, and p63. Similarly, they are negative for CK20, SMA, MSA, and S100. However, p63 is an efficient marker used to differentiate acinic cell carcinomas from mucoepidermoid carcinomas. On the other hand, myoepithelial carcinomas are malignant tumors formed in salivary glands with high myoepithelial differentiation. These tumor cells are diverse, and include spindled, stellate, epithelioid and clear cells with some resemblance to sarcoma, melanoma or other tumors (Figure 1). Immunoreactivity for both types of keratins and a minimum of one myoepithelial marker reaction is required to diagnose such tumors. Myoepithelial carcinomas mainly express vimentin (100%), calponin (75–100%), S100 (82–100%), CAM 5.2 (89%), EMA (27%), and SMA (35–50%), which shows that some markers are more frequently expressed than others, while some are occasionally expressed. Calponin is sensitive to myoepithelial carcinomas and is considered to be the most specific marker for these carcinomas. However, a combination of markers, such as CK AE/13, CAM5.2, CK5/6, SMA, S100, and calponin, can be effective for accurate diagnosis [8]. Table 4 provides insights on works in the literature that have used a wide range of markers and shows the reaction of the patients to those markers.

#### 1.2.3. Polymorphous Adenocarcinomas

Polymorphous adenocarcinomas are low-grade malignant epithelial carcinomas. The formation of these tumors takes place mainly in the minor salivary glands with mild cytologic features. These tumors consist of diverse growth patterns in different areas such as lobular, papillary, cribriform, ductal, and tubular. Such tumors express CK AE1/3, CAM 5.2, EMA, 34BE12, p53, p63, vimentin, and S100 with an infrequent level of SMA [8]. It was observed that 96% of canalicular adenomas show weak and strong cytoplasmic staining for GFAP in the ductal and myoepithelial cells. GFAP is considered to be more helpful in distinguishing pleomorphic low-grade adenocarcinomas from pleomorphic adenomas [38].

#### 1.2.4. Salivary Duct Carcinomas

Salivary duct carcinomas are aggressive and malignant epithelial tumors that form from intralobular and interlobular excretory ducts. These tumors express AR, GCDFP-15, CK AE1/3, CK7, 34BE12, CEA, and EMA. However, they can occasionally be positive for ER, PR, and S-100. Ki-67 shows an increased expression of more than 25%. Androgen receptors occur more frequently in men compared to women in salivary duct carcinomas. A rate of 80% overexpression of HER2/neu and p53 for salivary duct carcinomas is connected to a weaker level of prognosis [8].

#### 1.2.5. Pleomorphic Adenomas

Pleomorphic adenomas (PMAs) are the most common benign salivary gland tumors with an occurrence rate of 86% [39]. They were first described in 1859 by Billroth [6]. Histologically, PMAs consist of myoepithelial and epithelial cells in different morphological patterns (Figure 1) [40]. A PMA can develop into a benign but metastasizing subtype, the metastasizing pleomorphic adenoma (MPA), or a malignant mixed tumor called carcinoma ex-pleomorphic adenoma and carcinosarcoma. The probability of malignant transformation rises with longer existence of the tumor and increased patient age [41,42]. PMAs react mostly positively to calponin, CD9, GFAP, Mcl-2, NM23, p63, S-100, SMA, and Sox10. PLAG1 is specific for PMAs and is therefore widely used as diagnostic marker. Amylase, DOG1, HMGA-2, KIT, and MYB are sometimes positive. The usually negatively reacting markers are carbonic anhydrase VI, LPLUNC1, SPLUNC1, and SPLUNC2. Table 5 lists the relevant studies including the sample sizes of PMAs, the marker used, and the reactions.

## 2. State-of-the-Art Methods in Pre-Operative Diagnosis of Salivary Gland Tumors

Salivary gland tumors are rare and account for about 2% to 6.5% of all head and neck neoplasms. The rarity and morphological variations make them difficult to diagnose. Therefore, state-of-the-art diagnostic imaging methods have been adopted over the years for better differential diagnosis. Imaging provides crucial information for accurate localization of salivary gland tumors (e.g., localization in superficial and deep lobes) and differentiation between benignancy and malignancy and plays an important role in the staging procedure of salivary gland cancer. In addition, imaging also allows for differentiating recurrent malignant tumors from post-treatment changes and monitoring patient health after therapy. Magnetic resonance (MR) imaging is the most preferred modality for the evaluation of salivary gland tumors. Routine pre-contrast MR imaging plays an important role in accurate localization and assessment of locoregional extension of salivary gland tumors. Contrast-based MR imaging assesses the perineural spread of salivary cancer malignancy, which commonly occurs in patients with adenoid cystic carcinomas. Advanced MR imaging procedures such as diffusion-weighted MR imaging and dynamic contrast-enhanced (DCE) imaging effectively characterize particular salivary gland tumors. MR imaging is also practical for differentiating recurrence from post-treatment changes. Similarly, routine post-contrast computed tomography (CT) scans help to evaluate skull base cortical invasions, whereas PET-CT effectively detects distant metastases of salivary gland cancer [44].

It is essential to identify whether mass stems from the deep or superficial lobe of the parotid gland, as the surgical approaches for these tumors vary. Therefore, tumor localization is of utmost importance for accurate diagnosis. Another crucial aspect of diagnosis is identifying the intraparotidal pathway of the facial nerve, which is performed using anatomical landmarks. The three-dimensional cross-sectional imaging technique proposed by Atkinson highlights how the proposed approach can help demonstrate the facial nerve. A T2-weighted MR sequence can distinguish between ducts and nerves of low signal intensity. Generally, pleomorphic adenomas have a progressive radiocontrast agent enhancement, whereas Warthin tumors appear to be washed out. These tumors require significant enhancement, which is achievable using DCE and ADC MR imaging techniques. Similarly, T1- and T2-weighted imaging and contrast-enhanced imaging are used to identify the signal intensities and densities based on MR and CT imaging. Myoepitheliomas appear to be small and round tumors with smooth contours [44]. 

## 3. Fine Needle Aspiration of Salivary Gland Tumors

Fine needle aspiration (FNA) of salivary gland tumors is an important diagnostic tool for preoperative risk stratification. It helps to distinguish benign from malignant tumors and is, therefore, indispensable for therapeutic decisions [45]. FNA also helps to limit the need for surgical intervention and can thereby reduce treatment costs [46]. However, FNA has a lower sensitivity with high heterogeneity; therefore, it is advised to use as an additional tool for the detection of malignancies [45,47]. For example, in PMA, at least 25% of the aspirate smear consists of fibrillary stroma with intermixed myoepithelial cells surrounding gland ductal structures [48]. As MECs are very heterogeneous, diagnosis by FNA is difficult. Depending on the aspirated portion, false diagnoses such as abscesses or PMA may occur. To confirm a low-grade MEC by FNA, cells such as intermediate cells, mucin-secreting cells, and squamous cells must be present [49]. ACC often presents as normal acinar cells in FNA and is, thus, underdiagnosed. However, if the tumor is of predominant papillary architecture type, it resembles adenocarcinoma. Typically, acinar-cell-like tumor cells are seen in the FNA of ACCs, taking over their pattern. In some cases, one also sees capillary plexus or even papillary formations around a fibrovascular core [50]. In summary, FNA is suitable to diagnose homogeneous tumors with a high probability. For heterogeneous tumors such as salivary gland tumors, FNA is not suitable as the sole diagnostic tool, but can be used as an additional one.

## 4. Translocations

Translocations are considered to account for about 20% of all forms of cancers. In recent years, translocations and their resultant fusion oncogenes were found to be rare for epithelial tumors. These are genetic aberrations that are mainly found in hematolymphoid and soft-tissue neoplasms and are rare in other tumors. Difficulty arises in finding epithelial tumors with translocations because of growing carcinoma or acquiring the karyotypes of these tumors. It has recently been found that multiple salivary cancers comprise translocations, namely MEC, ACC, and other forms of carcinomas such as MASC and HCCC, among several more. Translocation emergence detection is more feasible with next-generation molecular markers, gene fusions, and whole gene sequencing. Salivary glands are appropriate for finding translocations as they show low-grade malignancy with relative homogeneity. Translocations become essential as every tumor with a specific subtype looks histologically similar. At the same time, they also appear different from typical salivary gland elements. Certain tumors with a complex nature of genomic abnormalities occur in the background of preneoplastic dysplasias. Considering these factors, it can be highlighted that tumors such as epithelial-myoepithelial carcinomas and basal cell adenocarcinomas may have translocations [51].

### 4.1. Translocations in Pleomorphic Adenomas (PAs)

Pleomorphic adenomas (PAs) comprise a myxoid matrix, chondroid, and other elements related to myoepithelial cells. These cells are often in the forms of epithelioid, spindled, plasmacytoid, and clear cells. Pleomorphic adenomas are known to conceal translocations that involve PLAG1 or HMGA2 in most cases. Such tumors have high morbidity rates and are subject to frequent recurrence if not resected promptly with wide safety margins. These tumors are an aggressive form of malignant salivary gland tumors with a significantly increased mortality rate. Abnormalities in PA occur in the 8q12 region, while some have balanced reciprocal translocations, including PLAG1, and some show wide variations in abnormalities with various fusions. The most commonly known fusion partner is CTNNB1, which is the gene coding for beta-catenin. PLAG1 upregulations are usually detected using immunohistochemistry but are not confined to those cases. Another fusion partner is the LIFR, which has similar upregulations as CTNNB1. Another minor known partner is SII that has been cloned. The ring chromosomes in specific PAs have been identified as fused with FGFR1 to PLAG1, which is one of the causes of similar-appearing tumors. PAs with numerous aberrations and the ones without any identifiable aberrations usually have a similar appearance. This suggests that FISH procedures for such gene types have limited diagnostic utility without prognostic significance [51].

### 4.2. Translocations in Mucoepidermoid Carcinomas (MECs)

MECs are some of the most common salivary gland carcinomas. They can be found in major or minor salivary glands. These tumors are also likely to occur in other locations such as the tracheobronchial tree, cervix, skin, or the upper respiratory tract. MECs have large ducts and comprise three cell types: mucinous cells, epidermoid cells, and intermediate cells. Typically, the occurrence of MECs should alert the clinicians to perform additional diagnostic tests. These tumors can mimic the mucinous components or have cystic variations. There were no prior diagnostic or prognostic markers until CRTC1 and MAML2 fusion came into use. This fusion is often used for prognosis, and about 55% of cases tended to have a positive outcome. Fusion-positive cases are more likely to be observed in younger patients with decreased recurrence rates. MAMLM2 FISH has been used for higher-grade translocation-positive MECs for prognostic purposes in some studies [51]. Another study has pointed out that 46% of high-grade MECs are found to be translocation-positive with minimal anaplasia and were high grade due to additional factors. On the contrary, high-grade translocation-negative MECs appear to have significant anaplasias with frequent mucinous differentiation. The MAML2 FISH fusion provides better outcomes, but the results are less convincing [52]. 

### 4.3. Translocations in Adenoid Cystic Carcinomas (ACCs)

ACCs are the most common form of salivary gland carcinoma after MECs. They are of intercalated duct origin, which signifies that there is a participation of both ductal and myoepithelial cells. ACCs comprise a combination of tubular and cribriform elements with bilayers and visible myoepithelial layers on the outer region. Recent studies include t (6;9) (q22–23; p23–24) in various karyotypes of ACCs. Cloning of the fusion of MYB oncogene and NFIB transcription factor gene has been tested for salivary glands, head and neck, and breast sites. However, MYB activation is required for fusion-negative cases. MYB overexpression shown with RT-PCR appeared more in fusion-positive cases and was confirmed by immunohistochemistry for the MYB protein. MYB protein expression tended to be negative or focal in non-ACC salivary cases. It has been noted that MYB expression in specific ACC cases has been strong but not very sensitive overall.

On the other hand, MYB has been found to play a crucial role in ACCs irrespective of the type of fusion. Additionally, MYB RNA has appeared to be fusion-positive for ACCs in all anatomic sites. Nevertheless, it is unclear whether translocation in ACC can be a useful diagnostic marker [51]. Table 6 offers an overview of translocation types and prevalence assessed in various studies.

## 5. Analysis of Markers

In recent years, many new immunohistochemical and molecular markers have been proposed for diagnosis, some of them being successful in distinguishing between the many different entities. In these papers, following markers proved to be helpful: carbonic anhydrase VI is only positive in ACC, PLAG1 in PMA, S-100 and SOX10 in PMA and ADCC, and LPLUNC and SPLUNC2 in MEC. Figure 2, Figure 3, Figure 4 and Figure 5 depict the possible markers that have made it easy to differentiate between tumors.

## 6. Future Directives 

Salivary gland tumors are complex due to the variations in their morphological characteristics. In order to achieve accurate detection and dose delivery, there is a need for the combination of emerging biological markers and next-generation diagnostic technologies. Furthermore, several types of research have highlighted the importance of saliva becoming a significant source of insights on possible diagnosis and prognosis. 

In the newest edition of the WHO Classification of Head and Neck Tumors from 2017, the increasing importance of translocation diagnostics is underlined. Regarding the selected tumors in this paper, eight molecular alterations are mentionable: PMA has a PLAG1 fusion in over 50% of cases and an HMGA2 fusion in approximately 15%. MECs possess a CRTC1-MAML2 alteration with a probability of 40–80% and a CRTC3-MAML2 alteration with around 5% probability. A CDKN2A deletion happens in 35% of MECs. MYB alterations are frequent in 80% of ADCC cases, followed by MYBL1 in 10% and NOTCH1 in 5–10% [41].

At present, there are about 100 types of salivary biomarkers that currently exist, ranging from non-organic compound biomarkers (sodium, calcium); protein biomarkers (p53, alpha-amylase); and DNA-, RNA-, and microRNA-related biomarkers (p53 gene codon 63, miR-125a) to metabolomics biomarkers (valine and lactic acid) and other miscellaneous biomarkers. Salivary-based biomarkers have gained importance for assessing the risk of malignant diseases. DNA- or RNA-based methods help to identify the tumors without traditional approaches. PCR identification techniques ensure accurate measurement, and new techniques such as saliva-based identification and salivary epithelial cells are being developed to improve accurate diagnosis. Salivary epithelial cells are known to secrete proteins in the bloodstream. Hence, they are considered a potential marker for the target site. Emerging salivary gland markers include molecular and protein-based markers that can provide critical information on the disease status. The increasing number of cases of oral cancer in recent years has led to research in salivary biomarkers for oral cancers such as salivary gland carcinomas [53].

## 7. Conclusions

Molecular analysis is an emerging approach of sequencing that helps clinicians to characterize tumors effectively. Evidence of translocation by molecular markers is not essential for diagnosis. However, the number of molecular alterations can refine the diagnosis. Use of the present molecular analysis and translocation needs to become more established in daily clinical life. On the other hand, technological advancements have introduced powerful diagnostic imaging techniques that can identify and aid in further prognosis of the disease and classification according to the malignancies. Thus, in time, several novel molecular alterations are expected be discovered, which might potentially increase the importance of markers.

## Figures and Tables

**Figure 1 ijms-22-06771-f001:**
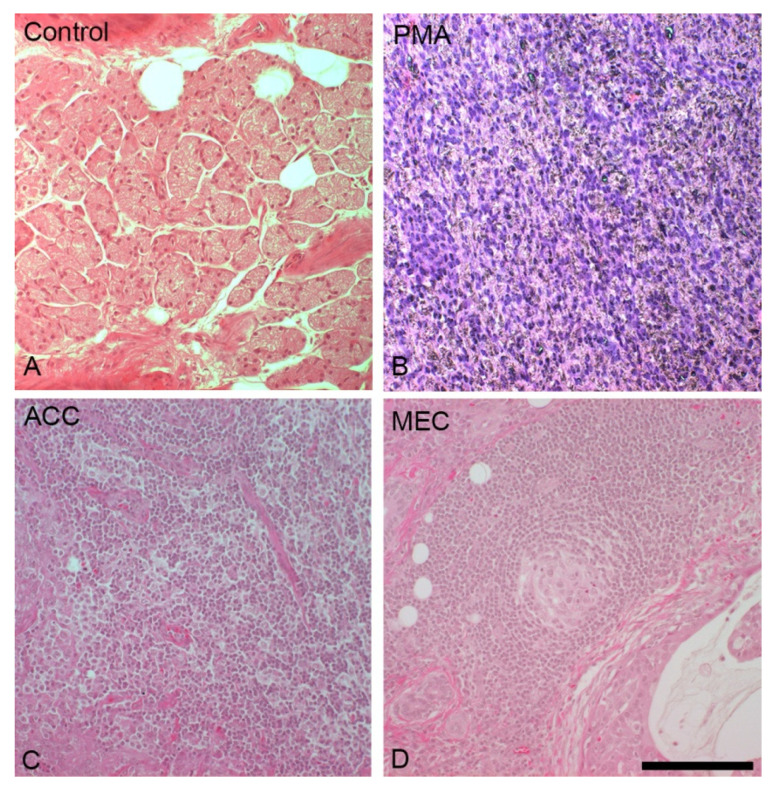
Comparison of the structural differences between control, pleomorphic adenoma (PMA), mucoepidermoid carcinoma (MEC), and acinar cell carcinoma (ACC) parotid gland tissue stained with HE staining. Control parotid tissue (**A**) is compared with PMA, which is composed of different cells of epithelial and mesenchymal lineage differentiations (**B**). ACC tumor cells resemble acinar cells in structure and pattern (**C**). MEC consists mostly of squamous epithelium and mucus-forming epithelium (**D**). Bars represent (**A**–**D**) 100 μm.

**Figure 2 ijms-22-06771-f002:**
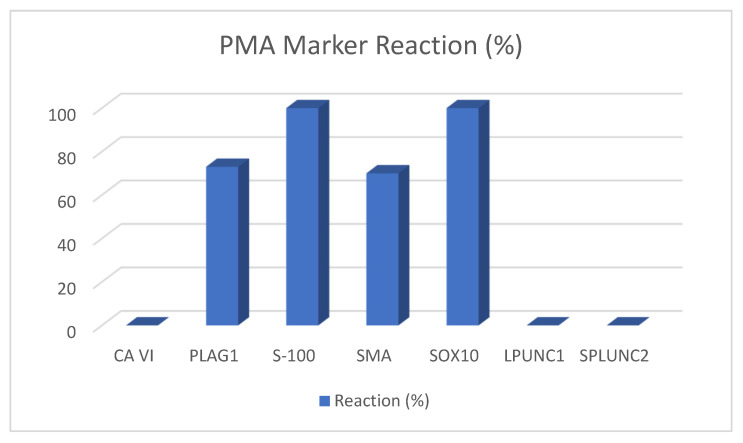
PMA marker reaction.

**Figure 3 ijms-22-06771-f003:**
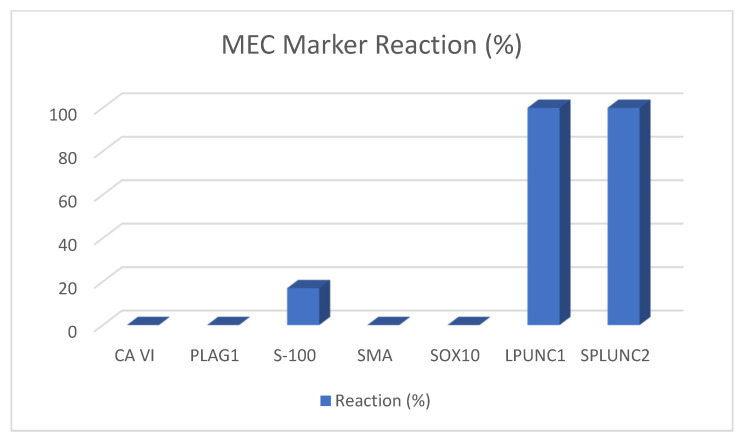
MEC marker reaction.

**Figure 4 ijms-22-06771-f004:**
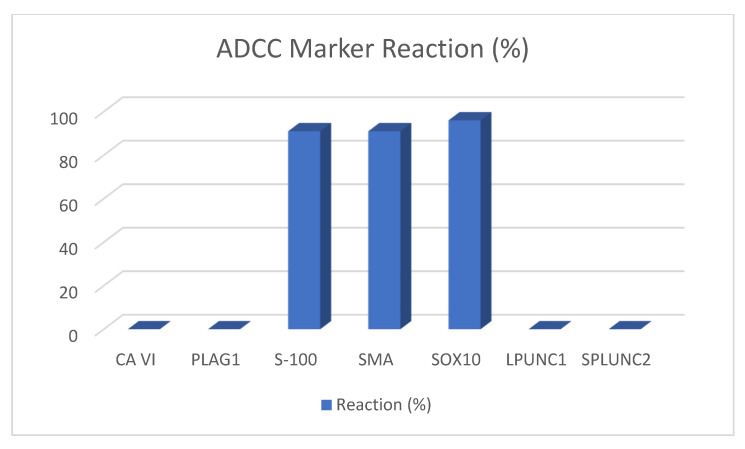
ADCC marker reaction.

**Figure 5 ijms-22-06771-f005:**
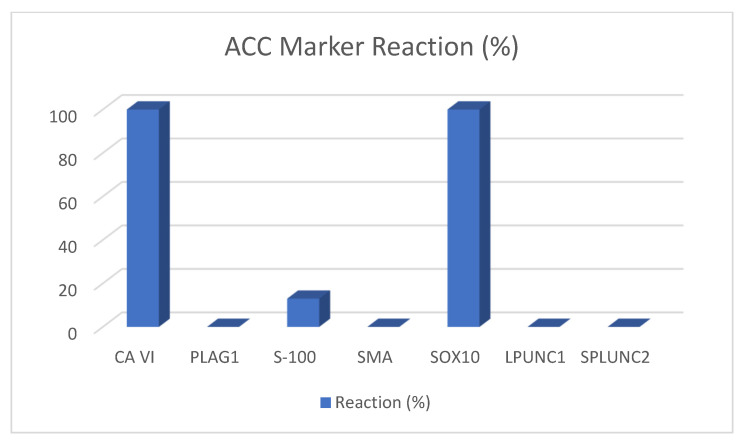
ACC marker reaction.

**Table 1 ijms-22-06771-t001:** Normal salivary gland.

Normal Salivary Gland	Marker	Reaction
General	Androgen receptor	Negative
	EBER in situ hybridization	Negative
	Ki-67 (MIB-1)	Few cells positive
General	Melan A	Negative
	P53	Negative
General	Renal cell carcinoma/CD10	Negative
General	S-100	Variable
General	CEA	Positive
Acinar cells	CK14	Positive
Abluminal cells	P63	Positive
Abluminal cells	CK, AE1/AE3	Positive
Myoepithelial cells	Calponin	Positive
Myoepithelial cells	GFAP	Positive (variable)
Myoepithelial cells	MSA	Positive
Myoepithelial cells	SMA	Positive
Myoepithelial cells	Vimentin	Positive
Luminal cells	EMA	Positive
	Gross cystic disease fluid protein	Positive
	CK, AE1/AE3	Positive
	HER2	Negative to weakly positive
Striated duct cells	Mitochondrial	Positive
	Alpha-amylase	Positive

**Table 2 ijms-22-06771-t002:** Adenoid cell carcinomas.

Study	Sample Size ADCC	Marker	Reaction	in %
[11]	Three of upper aerodigestive tract origin	Apocrine	0 of 3	0%
[11]	Three of upper aerodigestive tract origin	Caldesmon	1 of 3	33%
[11,12]	23 of salivary gland and head and neck origin; three of upper aerodigestive tract origin	Calponin	18 of 23; 3 of 3	78%; 100%
[13]	Five	Carbohydrate Ag 19-9	0 of 5	0%
[14]	Six of salivary gland origin	Carbonic anhydrase VI (CA6)	0 of 6	0%
[13]	Five	CEA	0 of 5	0%
[15]	Three of parotid gland origin	CD9	0 of 3	0%
[16]	Four of parotid origin	CD43	4 of 4	100%
[13]	Five	CEA	0 of 5	0%
[11]	Three of upper aerodigestive tract origin	CK7	3 of 3	100%
[11]	Three of upper aerodigestive tract origin	CK8	3 of 3	100%
[11]	Three of upper aerodigestive tract origin	CK14	3 of 3	100%
[11]	Three of upper aerodigestive tract origin	CK17	3 of 3	100%
[11]	Three of upper aerodigestive tract origin	CK19	3 of 3	100%
[11]	Three of upper aerodigestive tract origin	CK20	0 of 3	0%
[17]	15 of salivary gland origin	C-Kit	15 of 15	100%
[18]	24 of salivary gland origin	DOG-1	17 of 24	70%
[12]	23 of salivary gland and head and neck origin	GFAP	5 of 23	21%
[19]	Nine of salivary gland origin or potential mimickers	HMGA-2	0 of 9	0%
[19,20,21]	11 of salivary gland origin or potential mimickers; 66 of the head and neck area and breast; 37 of salivary gland origin	KIT	4 of 11; 62 of 66 (H300 (54 of 66)/A4502 (58 of 66)); 37 of 37	36%; 94%; 100%
[22]	13 of salivary gland origin	LPLUNC1	0 of 13	0%
[23,24]	14 of salivary gland origin; 25 of salivary gland origin	Maspin	12 of 14; 19 of 25	86%; 76%
[25]	13 of parotid origin	Mcl-1	12 of 13	92%
[23]	14 of salivary gland origin	MCM2	12 of 14	86%
[11]	Three of upper aerodigestive tract origin	Mit	2 of 3	66%
[19,21]	Nine; 37 of salivary gland origin	MYB	4 of 11; 24 of 37	36%; 65%
[26]	Six of upper aerodigestive tract and lung origin, 14 of salivary gland origin, one of maxilla origin	NM23		68,70%
[12,27]	23 of salivary gland and head and neck origin; 16 of salivary gland	p63	21 of 23; 16 of 16	91%; 100%
[27]	16 of salivary gland	p73	16 of 16	100%
[19]	Eleven of salivary gland origin or potential mimickers	PLAG1	0 of 11	0%
[12]	23 of salivary gland and head and neck origin	S-100	21 of 23	91%
[11,12]	23 of salivary gland and head and neck origin; three of upper aerodigestive tract origin	SMA	21 of 23; 3 of 3	91%; 100%
[11]	Three of upper aerodigestive tract origin	SMM	2 of 3	66%
[22]	13 of salivary gland origin	SPLUNC1	0 of 13	0%
[22]	13 of salivary gland origin	SPLUNC2	0 of 13	0%
[28]	28	SOX4	28 of 28	100%
[12]	23 of salivary gland and head and neck origin	SOX10	22 of 23	96%

**Table 4 ijms-22-06771-t004:** Results of patient probes with mucoepidermoid carcinomas to various markers.

Study	Sample Size MEC	Marker	Reaction	in %
[31]	Five of parotid origin	Amylase	0 of 5	0%
[34]	59 of upper aerodigestive tract origin	Androgen receptor	0 of 59	0%
[11]	10 of upper aerodigestive tract origin	Apocrine	0 of 10	0%
[35]	173 of upper aerodigestive tract origin	Bcl-2		63%
[11]	10 of upper aerodigestive tract origin	Caldesmon	0 of 10	0%
[11,12]	Six; 10 of salivary gland and head and neck origin	Calponin	0 of 6; 1 of 10	0%; 10%
[13]	Seven	Carbohydrate Ag 19-9	4 of 7	57.10%
[14]	Five of salivary gland origin	Carbonic anhydrase VI (CA6)	0 of 5	0%
[13,35]	173 of upper aerodigestive tract origin	CEA	2 of 7	28.57%; 68.6%
[35]	173 of upper aerodigestive tract origin	C-erb-2		80%
[15]	Six of parotid origin	CD9	5 of 6	83%
[34]	59 of upper aerodigestive tract origin	CK5/6	60 of 64	93.75%
[11]	10 of upper aerodigestive tract origin	CK7	10 of 10	100%
[11]	10 of upper aerodigestive tract origin	CK 8	10 of 10	100%
[11]	10 of upper aerodigestive tract origin	CK14	10 of 10	100%
[11]	10 of upper aerodigestive tract origin	CK 17	10 of 10	100%
[11]	10 of upper aerodigestive tract origin	CK19	10 of 10	100%
[11]	10 of upper aerodigestive tract origin	CK20	1 of 10	10%
[36]	11 of upper aerodigestive tract origin	Cox-2	11 of 11	100%
[18]	Eight of salivary gland origin	DOG-1	3 of 8	37.50%
[12]	Six of salivary gland and head and neck origin	GFAP	2 of 6	33%
[34]	71 of upper aerodigestive tract origin	Her2/neu	26 of 71	37%
[19]	Three of salivary gland origin or potential mimickers	HMGA-2	0 of 3	0%
[20,21]	Nine of head and neck origin; 23 of upper aerodigestive tract origin	KIT (CD117)	H300 (1 of 9)/A4502 (0 of 9); 10 of 23	1%; 43%
[22]	10 of salivary gland origin	LPLUNC1	10 of 10	100%
[23,24]	15 of salivary gland origin; 15 of salivary gland origin	Maspin	15 of 15; 13 of 15	100%; 86.7%
[25]	12 of parotid origin	Mcl-2	11 of 12	92%
[23]	15 of salivary gland origin	MCM2	15 of 15	100%
[11]	10 of upper aerodigestive tract origin	Mit	10 of 10	100%
[37]	40 of salivary gland origin	MUC1	40 of 40	100%
[37]	40 of salivary gland origin	MUC2	2 of 40	5%
[37]	40 of salivary gland origin	MUC4	38 of 40	95%
[37]	40 of salivary gland origin	MUC5AC	29 of 40	72%
[37]	40 of salivary gland origin	MUC5B	33 of 40	82%
[37]	40 of salivary gland origin	MUC6	13 of 40	32%
[37]	40 of salivary gland origin	MUC7	2 of 40	5%
[19,21]	23 of the upper aerodigestive tract; three of salivary gland origin or potential mimickers	MYB	1 of 23; 1 of 3	4%; 33%
[26]	Seven of upper aerodigestive tract, nine of salivary gland origin, one of maxilla, and two of mandibula origin	NM23	18 of 19	92.80%
[13,35]	Seven; seven of upper aerodigestive tract origin	p53	4 of 7	57.1%; 16.4%
[12,27,34]	Six of salivary gland and head and neck origin; four; 65 of upper aerodigestive tract origin	p63	2 of 6; 4 of 4; 62 of 65	33%; 100%; 95.3%
[27]	Four	p73	4 of 4	100%
[35]	No sample size number available	PCNA		92.9%
[19]	Three of salivary gland origin or potential mimickers	PLAG1	0 of 3	0%
[12]	Six of salivary gland and head and neck origin	S-100	1 of 6	17%
[11,12]	10 of upper aerodigestive tract origin; six of salivary gland and head and neck origin	SMA	0 of 10; 0/6	0%
[11]	10 of upper aerodigestive tract origin	SMM	0 of 10	0%
[22]	10 of salivary gland origin	SPLUNC1	0 of 10	0%
[22]	10 of salivary gland origin	SPLUNC2	10 of 10	100%
[12]	Six of salivary gland and head and neck origin	SOX10	0 of 6	0%

**Table 5 ijms-22-06771-t005:** Pleomorphic adenoma result analysis.

Study	Sample Size PMA	Marker	Reaction
[31]	10 of parotid origin	Amylase	10% (1 of 10)
[12]	10 (thereof four carcinomas)	Calponin	70% (7 of 10)
[19,20,21]	30; 16; 16 of salivary gland origin	KIT	10%; 19%; 44%
[14]	Five of salivary gland origin	Carbonic anhydrate VI	Zero (0 of 5)
[15]	16 of parotid origin	CD9	68.75% (11 of 16)
[18]	14 of salivary gland origin	DOG1	7% (1 of 14)
[22]	10 of salivary gland origin	SPLUNC1	0% (0 of 10)
[22]	10 of salivary gland origin	SPLUNC2	0% (0 of 10)
[22]	10 of salivary gland origin	LPLUNC1	0% (0 of 10)
[12]	10 of salivary gland and head and neck origin (thereof four carcinomas)	GFAP	90% (9 of 10)
[26]	16	NM23	75%
[12]	10 of salivary gland and head and neck origin (thereof four carcinomas)	P63	80% (8 of 10)
[25]	30 of parotid origin	Mcl-2	73% (22 of 30)
[12]	10 of salivary gland and head and neck origin (thereof four carcinomas)	S100	100% (10 of 10)
[12]	10 of salivary gland and head and neck origin (thereof four carcinomas)	SMA	70% (7 of 10)
[19,21]	29 of salivary gland origin or potential mimickers; 16 of salivary gland origin	MYB	4% (1 of 4); 6% (1 of 16)
[19,43]	30 of salivary gland origin or potential mimickers; 45 of salivary gland origin	PLAG1	73% (22 of 30); 100% (45 of 45)

**Table 6 ijms-22-06771-t006:** Overview of translocation types and prevalence.

Tumor	Gene	Chromosomal Rearrangement	Prevalence
PMA	PLAG1 fusions	8q12 translocations	>50%
	HMGA2 fusions	12q13–15 translocations	~15%
MEC	CRTC1-MAML2	t (11; 19) (q21; p13)	40–80%
	CRTC3-MAML2	t (11; 15) (q21; q26)	~5%
	CDKN2A deletion	9p21.3	~35%
ADCC	MYB fusion/activation	6q22–23 translocations	~80%
	MYBL1 fusion/activation	8q13 translocations	~10%
	NOTCH1 mutation	-	5–10%

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
