# Peer review of "2021 Update on Diagnostic Markers and Translocation in Salivary Gland Tumors"

_ijms, 2021, doi:10.3390/ijms22136771_

Round 1

Reviewer 1 Report

This is an extenseive review on markers ans translocations in salivary gland tumors. the paper is a bit too long and some chapters may be shortened or deleted.

for example chapter 2 does not add useful information and should be deleted.

chapter 3: differential diagnosis should be named pre-operative diagnosis. Diagnostic identification includes of course MRI but cytopathology after FNA is able to indicate in most cases the presence of a neoplasm, its type and/or a tumor diagnosis. This should be mentioned.

3.1. Radiotherapy chapter should be removed and replaced by FNA

Author Response

Point to point answers to reviewer comments

We would like to thank the reviewers for reviewing the manuscript with constructive and supportive criticism and the important suggestions to further improve the manuscript. Within the revised version of the manuscript, we addressed the comments of the reviewers in a point by point fashion. These changes were appropriately incorporated and highlighted within the revised manuscript. We hope that the current revised version is improved and more appropriate for publication in the special issue “Recent Advances in Salivary Gland and Their Function” in International Journal of Molecular Sciences.

.
Reviewer #1: This is an extensive review on markers ans translocations in salivary gland tumors. the paper is a bit too long and some chapters may be shortened or deleted.

for example chapter 2 does not add useful information and should be deleted.

We would like to thank the reviewer for this observation. As suggested we deleted chapter 2.

chapter 3: differential diagnosis should be named pre-operative diagnosis.

Thank you for this observation and comment. As suggested we changed the name of the chapter.

Diagnostic identification includes of course MRI but cytopathology after FNA is able to indicate in most cases the presence of a neoplasm, its type and/or a tumor diagnosis. This should be mentioned.

3.1. Radiotherapy chapter should be removed and replaced by FNA

We would like to thank the reviewer for evaluation and the comment. As suggested, we have deleted the chapter and replaced it with one about FNA.

Reviewer 2 Report

Thank you for your paper. I've found it very interesting, well organized, well written. I suggest, if possible and if the Editor inChief agrees, to add some istological picture and immunohistochemical too (obviously, the most frequently used), as to complete the paper well. Also, I suggest to briefly discuss the diagnostic iter too (FNA) in order to complet all the aspect of salivary gland neplasm approach. Finally, in the MEC section should be cited  the initial form of the tumour as recently suggested in literature. Capodiferro S, Ingravallo G, Limongelli L, Mastropasqua MG, Tempesta A, Favia G, Maiorano E. Intra-Cystic (In Situ) Mucoepidermoid Carcinoma: A Clinico-Pathological Study of 14 Cases. J Clin Med. 2020 Apr 18;9(4):1157. doi: 10.3390/jcm9041157. PMID: 32325647; PMCID: PMC7231055.

Thank you again for your paper 

Author Response

Point to point answers to reviewer comments

We would like to thank the reviewers for reviewing the manuscript with constructive and supportive criticism and the important suggestions to further improve the manuscript. Within the revised version of the manuscript, we addressed the comments of the reviewers in a point by point fashion. These changes were appropriately incorporated and highlighted within the revised manuscript. We hope that the current revised version is improved and more appropriate for publication in the special issue “Recent Advances in Salivary Gland and Their Function” in International Journal of Molecular Sciences.

Reviewer #2: Thank you for your paper. I've found it very interesting, well organized, well written. I suggest, if possible and if the Editor inChief agrees, to add some istological picture and immunohistochemical too (obviously, the most frequently used), as to complete the paper well.

Also, I suggest to briefly discuss the diagnostic iter too (FNA) in order to complet all the aspect of salivary gland neplasm approach.

We would like to thank the reviewer for this observation. We added a chapter about FNA and histological pictures.

Finally, in the MEC section should be cited  the initial form of the tumour as recently suggested in literature. Capodiferro S, Ingravallo G, Limongelli L, Mastropasqua MG, Tempesta A, Favia G, Maiorano E. Intra-Cystic (In Situ) Mucoepidermoid Carcinoma: A Clinico-Pathological Study of 14 Cases. J Clin Med. 2020 Apr 18;9(4):1157. doi: 10.3390/jcm9041157. PMID: 32325647; PMCID: PMC7231055.

Thank you for this observation and comment. We have gone through the suggested manuscript and added necessary information.